# Comprehensive Effects of N Reduction Combined with Biostimulants on N Use Efficiency and Yield of the Winter Wheat–Summer Maize Rotation System

**Junji Li** [1]**, Haiyan Ma** [1]**, Hongliang Ma** [1]**, Fang Lei** [1]**, Dahai He** [1]**, Xiulan Huang** [1]**, Hongkun Yang** [1] **and Gaoqiong Fan** [1,2,3,*]

1    Crop Ecophysiology and Cultivation Key Laboratory of Sichuan Province, Sichuan Agricultural University, Chengdu 611130, China; 2021301086@stu.sicau.edu.cn (J.L.); 2020101010@stu.sicau.edu.cn (H.M.); 2019101007@stu.sicau.edu.cn (H.M.)
2    Key Laboratory of Crop Eco-Physiology & Farming System in Southwest China, Ministry of Agriculture and Rural Affairs, Chengdu 611130, China
3    State Key Laboratory of Crop Gene Exploration and Utilization in Southwest China, Ministry of Science and Technology, Chengdu 611130, China
*    Correspondence: fangao20056@126.com

**Abstract:** Biostimulants sprayed exogenously boost crop yield and quality. However, the effects of the co-application of biostimulants and fertilizers as base fertilizers in soil are still uncertain. The objective of this study was to investigate the overall effects of reducing N levels in conjunction with the application of biostimulants on the yield formation and N utilization of wheat and maize. Therefore, based on the winter wheat–summer maize rotation system in the modern R&D base of Sichuan Agricultural University, soil enzyme activities, soil inorganic nitrogen dynamic content, crop nitrogen accumulation and transportation, crop yields, and composition were determined. To achieve this, a total of nine treatments were established based on the winter wheat–summer maize rotation system. The experiment included the following treatments: no fertilization (CK0); one-time application of common compound fertilizer (CK1, applied at a rate of 225 kg ha$^{-1}$); common compound fertilizer as base fertilizer + urea as topdressing fertilizer (CK2, applied at a rate of 225 kg ha$^{-1}$, base/topdressing, 6/4); biostimulant + common compound fertilizer with 20% or 30% N reduction (jf-20%, jf-30%); biostimulant chelated urea-formaldehyde fertilizer reducing N by 20%, 30%, or 40% (jn-20%, jn-30%, or jn-40%); and biostimulant chelated urea-formaldehyde fertilizer reducing N by 40% and combined with organic fertilizer, thereby totally reducing N by 27% (jny-27%). The results demonstrated that the application of a biostimulant increased the activities of urease, nitrate reductase, and nitrite reductase in the soil of wheat and maize during the flowering stage. At the same time, the amount of residual nitrate and ammonium N in the soil at maturity was reduced. Furthermore, when N application was appropriately reduced, wheat and maize plants treated with jf, jn, and jny showed a significant increase in N assimilation after the flowering stage, resulting in higher N accumulation in the grains at maturity and ultimately improving the yield compared to CK1 and CK2. The combined use of biostimulants also had a significant positive impact on N use efficiency (NUE). During the two-year period, the NUE in the wheat season showed an increase ranging from 6.70% to 24.00% compared to CK1 and from 5.30% to 22.60% compared to CK2. Similarly, in the maize season, the NUE increased by a range of 11.60% to 22.57% compared to CK1 and from 11.78% to 22.75% compared to CK2. Overall, biostimulants enhanced N absorption and transportation by matching crop N requirements in the mid-to-late stages and improved NUE and yield under appropriate N reduction. This study contributes to the design of improved measures for N reduction and yield stabilization in order to promote sustainable agricultural development.

**Keywords:** winter wheat–summer maize; biostimulant; nitrogen accumulation; yield; nitrogen use efficiency; soil enzyme activity

## 1. Introduction

Wheat–maize is a major rotation system in the world and plays a crucial role in food security. According to projections, global maize and wheat production will need to increase by 67% and 38%, respectively, by 2050 to meet the growing population's demands. However, the current increase rate is insufficient to reach this target [1,2]. Therefore, enhancing the yield of wheat and maize has become a crucial challenge for modern agricultural production. With the rising demand for food and the decreasing availability of arable land [3], agricultural producers often choose to increase the use of chemical fertilizers to achieve higher crop yields. However, this approach increases production costs and reduces the N utilization rate [4]. Moreover, excess N can lead to severe environmental issues, including soil degradation and groundwater contamination through nitrate infiltration [5–7].

Improving the efficiency of N use is crucial to sustainable agricultural production [8]. Slow-release fertilizers showed greater potential for increasing the NUE of crops, such as urea-formaldehyde. However, matching dynamic nutrition with the nutrient demand of crops is challenging [9–11]. Moreover, the high cost of slow-release fertilizers hinders their application in field production [12]. Therefore, organic fertilizer with slow-release fertilizers has also become common because of its renewable nature and environmental protection. It can enhance soil aggregation, increase soil organic carbon storage, and promote soil enzyme activity [13,14]. However, organic fertilizers have lower nutrient content and slower nutrient release rates than chemical fertilizers, which may not effectively meet production requirements [15,16]. Therefore, it is necessary to find a way to make up for the deficiency of the above fertilizer.

Biostimulants increase microbial activity, which enhances crop growth and productivity by improving environmental conditions and increasing tolerance to biotic or abiotic stresses [17,18]. Biostimulants derived from natural extracts include free amino acids, humus extracts, seaweed extracts, and chitin and its derivatives (such as chitosan) [19]. Among them, seaweed extracts and chitosan are extensively studied. Seaweed extracts are rich in hormones, amino acids, and other substances that significantly contribute to crop growth and development. Moreover, the polysaccharides and alginic acid in seaweed extracts act as chelating agents in the soil, facilitating the adsorption and slow release of nutrients and stabilizing the granular structure and colloid characteristics of the soil [20–22]. Chitosan and its derivatives are a type of natural polyamine glucose commonly derived from the shells of shrimps, crabs, and shellfish. They are widely used in agricultural production as plant growth regulators and soil conditioners [23,24], for example by increasing the levels of soluble proteins and other cold-resistant substances in plants, effectively combating low adversity in cold conditions. In addition, biostimulants can also enhance soil's physical and chemical properties, improve nutrient utilization efficiency, and promote plant growth through their exceptional adsorption capacity, film formation, and slow-release properties [25–27]. These results suggest the possible potential of reduced fertilization for biostimulants combined with fertilizers.

Currently, numerous studies have focused on the role of in vitro spraying of crops with biostimulants in increasing yield and improving quality [28,29]. However, the impact of their simultaneous application with fertilizers on soil nutrient release, N acquisition, and N utilization by crops remains unclear. This study aimed to investigate the combined effects of N and biostimulants on soil N content, soil enzyme activity, N acquisition, and utilization by plants in a winter wheat–summer maize rotation system. The findings will help find better solutions for improving annual crop yields and N utilization efficiency.

## 2. Materials and Methods

### 2.1. Site Description and Materials

The experiment was carried out in May 2021 and ended in May 2023, and the site was located at the Modern Agriculture Research and Development Base of Sichuan Agricultural University in Qiquan Town, Chongzhou City, Sichuan Province (30°53′25″ N,

103°63′43″ E). The location has a humid subtropical monsoon climate zone and sandy loam soil. The basic fertility was measured before sowing, the specific values are shown in Table 1, and the meteorological data are shown in Figure 1. The meteorological data for the crop growth period in each quarter have been separated by dotted lines in Figure 1, and the time for measuring basic fertility is the interval between maize harvest and wheat sowing.

**Table 1.** Soil fertility of the experiment site over two years.

| Years | pH | SOM (g kg⁻¹) | TN (g kg⁻¹) | $NH_4^+$-N (mg kg⁻¹) | $NO_3^-$-N (mg kg⁻¹) | AP (mg kg⁻¹) | AK (mg kg⁻¹) |
|---|---|---|---|---|---|---|---|
| 2021–2022 | 6.87 | 20.2 | 1.8 | 2.4 | 4.3 | 11.8 | 85.6 |
| 2022–2023 | 6.91 | 19.3 | 1.7 | 2.4 | 4.4 | 11.6 | 84.4 |

SOM: soil organic matter; TN: total N; $NH_4^+$-N: ammonium N; $NO_3^-$-N: nitrate N; AP: available phosphorus; AK: available potassium.

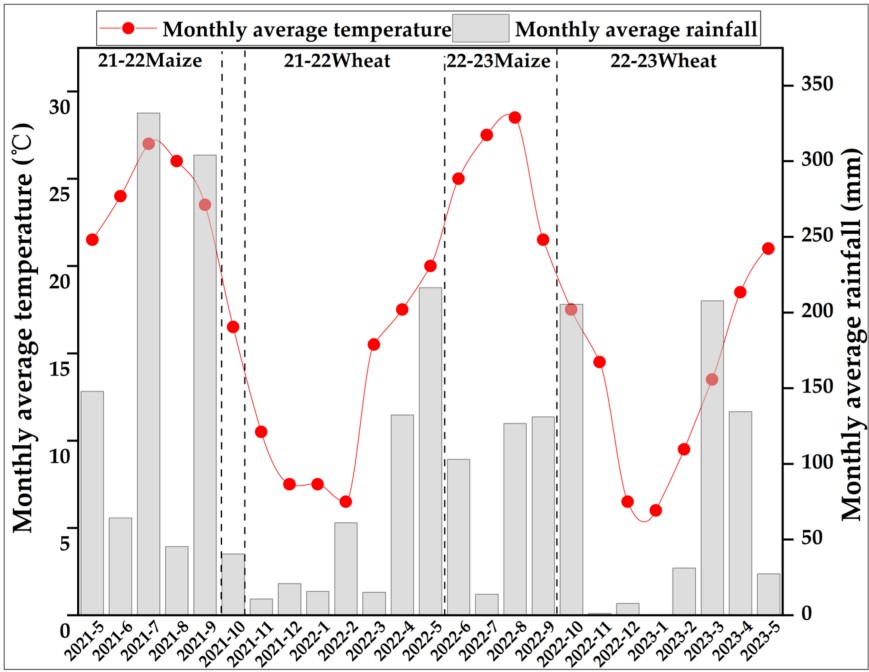

**Figure 1.** Meteorological data of the experiment site over two years.

The wheat variety used in the experiment was Shu Mai 133 and the maize variety was Zheng Hong No.6. The used fertilizer materials included biostimulants (homogeneous liquid) and biostimulant chelated urea-formaldehyde fertilizers (solid fertilizer with N:P:K ratio of 25:9:16). The biostimulant used in this experiment was a complex formed by polymerizing seaweed polysaccharide and chitosan with a nonionic active agent, which was obtained from Jijian Bio-technology Co., (Chengdu, China). The organic fertilizer had a nitrogen, phosphorus, and potassium ratio of 0.98:1:0.5. Common compound fertilizer had an N:P:K ratio of 26:10:16. Urea with an effective nitrogen content of 46%, organic fertilizer, common compound fertilizer, and urea were purchased from the local agricultural market.

### 2.2. Experimental Design and Management

The winter wheat–summer maize crop rotation model was implemented in this study, with the previous crop being wheat. Following the harvest, all straw was removed from the experimental plot to ensure that soil fertility was not compromised. The plot was then tilled and sorted. Each plot in the experiment had an area of 21 m² (4.2 × 5 m), and there were 3 replicates per treatment, resulting in a total of 27 plots. Wheat was sown in rows and covered with soil after sowing. The row spacing of each plot was 20 cm, with a total

of 21 rows. The initial seedling density was set at $2.0 \times 10^6$ plants ha$^{-1}$. Sowing dates for wheat were 27 October 2021 and 24 October 2022, with harvest dates of 8 May 2022 and 4 May 2023, respectively. The topdressing dates for CK2 were 24 November 2022 and 22 November 2023, respectively. For maize, it was sown in acupoints with a row spacing of 70 cm and a hole spacing of 25 cm. The initial seedling density at the four-leaf stage was $5.72 \times 10^4$ plants ha$^{-1}$. Maize was sown on 13 May 2021 and 19 May 2022, respectively, and harvested on 2 September 2021 and 11 September 2022, respectively. Except for urea as topdressing, other fertilizers were used as base fertilizers after crop seeding. The field management practices for wheat and maize were consistent with those followed in local high-yield fields. The study conducted a single-factor randomized block test consisting of nine treatments. These treatments were named as follows: CK0 (control, no fertilization), CK1 (control, one-time fertilization of common compound fertilizer with a conventional N application rate of 187.5 kg ha$^{-1}$ in wheat and 225 kg ha$^{-1}$ in maize), CK2 (comparison, which involved using common compound fertilizer as base fertilizer and urea topdressing). The N application amount was conventional, with 187.5 kg ha$^{-1}$ in wheat and 225 kg ha$^{-1}$ in maize. The ratio of basal fertilizer to topdressing fertilizer was 6:4. jf-20% and jf-30% (the biostimulant and common compound fertilizers were stirred and mixed; compared to the common N application rate, it was reduced by 20% and 30%, respectively); jn-20%, jn-30%, and jn-40% (the biostimulant chelated urea-formaldehyde fertilizer was used; compared with the common N application rate, it was reduced by 20%, 30%, and 40%, respectively); and jny-27% (jn-40% treatment was used and mixed with organic fertilizer, compared with the common N application rate, it was reduced by 27%). Except for topdressing, the fertilizer was applied once as the base fertilizer. The fertilization treatment and N reduction gradient are shown in Table 2.

**Table 2.** Fertilization treatment and N reduction gradient.

| Treatment | Wheat | | Maize | | N% (Compared to Common Compound Fertilizer) | Base/ Topdressing |
|---|---|---|---|---|---|---|
| | Fertilization Patterns and Gradients | N Dosage (kg ha$^{-1}$) | Fertilization Patterns and Gradients | N Dosage (kg ha$^{-1}$) | | |
| CK0 | No Fertilizer | 0.0 | No Fertilizer | 0.0 | −100 | — |
| CK1 | Common compound fertilizer (721 kg ha$^{-1}$) Without topdressing | 187.5 | Common compound fertilizer (865.5 kg·ha$^{-1}$) without topdressing | 225.0 | 0 | 1:0 |
| CK2 | Common compound fertilizer (432 kg ha$^{-1}$) + Urea (163 kg ha$^{-1}$) Seedling topdressing | 187.5 | Common compound fertilizer (519 kg·ha$^{-1}$) + Urea (195.6 kg·ha$^{-1}$) Belling stage topdressing | 225.0 | 0 | 6:4 |
| jf-20% | Biostimulant (3000 mL ha$^{-1}$) + Common compound fertilizer (577 kg·ha$^{-1}$) | 150.0 | Biostimulant (3000 mL ha$^{-1}$) + Common compound fertilizer (692 kg·ha$^{-1}$) | 180.0 | −20 | 1:0 |
| jf-30% | Biostimulant (3000 mL ha$^{-1}$) + Common compound fertilizer (505 kg·ha$^{-1}$) | 131.3 | Biostimulant (3000 mL ha$^{-1}$) + Common compound fertilizer (606 kg·ha$^{-1}$) | 157.5 | −30 | 1:0 |
| jn-20% | Biostimulant chelated urea-formaldehyde fertilizer (525 kg·ha$^{-1}$) | 150.0 | Biostimulant chelated urea-formaldehyde fertilizer (720 kg·ha$^{-1}$) | 180.0 | −20 | 1:0 |

**Table 2.** *Cont.*

| Treatment | Wheat | | Maize | | N% (Compared to Common Compound Fertilizer) | Base/ Topdressing |
|---|---|---|---|---|---|---|
| | Fertilization Patterns and Gradients | N Dosage (kg ha$^{-1}$) | Fertilization Patterns and Gradients | N Dosage (kg ha$^{-1}$) | | |
| jn-30% | Biostimulant chelated urea-formaldehyde fertilizer (450 kg·ha$^{-1}$) | 131.3 | Biostimulant chelated urea-formaldehyde fertilizer (630 kg·ha$^{-1}$) | 157.5 | −30 | 1:0 |
| jn-40% | Biostimulant chelated urea-formaldehyde fertilizer (450 kg·ha$^{-1}$) | 112.5 | Biostimulant chelated urea-formaldehyde fertilizer (540 kg·ha$^{-1}$) | 135.0 | −40 | 1:0 |
| jny-27% | Biostimulant chelated urea-formaldehyde fertilizer (450 kg·ha$^{-1}$) + Organic fertilizer (2500 kg·ha$^{-1}$) | 137.0 | Biostimulant chelated urea-formaldehyde fertilizer (540 kg·ha$^{-1}$) + Organic fertilizer (3000 kg·ha$^{-1}$) | 164.0 | −27 | 1:0 |

*2.3. Sampling and Measurements*

2.3.1. Soil N Content and Enzyme Activity

At the nodulation, flowering, and maturation stages of wheat and the flowering and maturation stages of maize, five points were randomly selected in each plot, and the soil layer of 0–20 cm was evenly obtained, mixed, and dried naturally, and then ground through a 20–60 mesh sieve for the determination of soil N and related enzyme activities. Nitrate N was determined by the potassium chloride leaching ultraviolet spectrophotometer method, ammonium N was determined by the potassium chloride leaching indigo phenol blue colorimetric method, soil urease activity was determined by the sodium phenol hypochlorite colorimetric method (expressed as $NH_4^+$-N produced in the culture solution within 24 h) [30], and soil nitrate reductase activity and soil nitrite reductase activity were determined by Soil Nitrate Reductase (NR) ELISA Research Kit and Soil Nitrite Reductase (NiR) ELISA Research Kit produced by Jiangsu Enzyme Immunity Industry Co.

2.3.2. Plant N Accumulation, Transportation, and Yield

During the flowering and maturity of wheat, 30 uniform samples of wheat plants were randomly retrieved from each plot, bagged in separate organs, and then dried at 105 °C for 30 min and then at 80 °C until constant weight. After being ground through a 60-mesh sieve and digested in concentrated $H_2SO_4$, the N content of the relevant parts was measured by the Kjeldahl N fixation method. In addition, before harvest, 1 m$^2$ of wheat plant samples were randomly selected in each plot to count the fertile spike numbers and the grain numbers of 30 spikes, and, finally, 3 m$^2$ of wheat samples were randomly harvested in each plot and threshed to calculate the yield according to the standard moisture content of 13% and to measure the thousand kernel weight. During the flowering and maturation stages of maize, five plants were selected and treated as wheat, and then the N content of each organ was measured. At harvest, the number of rows and kernels of 40 maize plants was continuously investigated, and the final threshing was carried out according to the standard moisture content of 14% to calculate the yield and measure the weight of 500 grains.

The calculation formula was as follows [31,32]:

N transportation in pre-flowering nutrient organs (stem and leaf) = N accumulation in nutrient organs at anthesis − N accumulation in nutrient organs at maturity

N transportation rate of pre-flowering nutrient organs = (N accumulation of nutrient organs at anthesis − N accumulation of nutrient organs at maturity)/N accumulation of nutrient organs at anthesis × 100%

Contribution of pre-flowering nutrient organ N translocation to seed N accumulation at maturity = (N accumulation in nutrient organs at flowering − N accumulation in nutrient organs at maturity)/N accumulation in grain N at maturity × 100%

N assimilation after flowering = N accumulation in grains at maturity − N translocation in nutrient organs before flowering

Contribution of post-flowering camp N assimilation to seed N accumulation at maturity = post-flowering N assimilation/seed N accumulation at maturity × 100%

N harvest index (NHI) = seed N accumulation/total plant N accumulation

NUE = (aboveground N uptake of plants in N application area − aboveground N uptake of plants in N non-application area)/N application × 100%

N partial productivity (NPP) = grain yield/N applied

Economic benefits = grain yield benefits − cost of production (fertilizer seed pesticide costs, land rent, field management costs, and seed and pesticide costs)

### 2.4. Date Analysis

The Excel2010 software was used to collate the experimental data; the IBM SPSS Statistics 22.0 software was used for variance analysis (ANOVA). The least significant difference (LSD) method was used for the significance test ($p < 0.05$). The image visualization was performed by Origin 2021.

## 3. Results

### 3.1. Soil N Supply Capacity

Under the combined treatments with biostimulants, soil nitrate N and ammonium N levels tended to decrease in the jointing, flowering, and maturation stages of wheat compared to CK1 and CK2. This decline was particularly significant at the maturation stage. In particular, the soil nitrate and nitrogen contents decreased in the first year by 14.90% to 23.40% and 12.3% to 21.1%, and in the second year by 9.9% to 18.6% and 9.3% to 18.0%. Similarly, at the maturation stage, the content of ammonium N in the soil decreased significantly by 16.5% to 20.0% and 16.5% to 20.1% in the first year and by 2.9% to 5.5% and 2.1% to 4.7% in the second year. At the maturation stage, maize levels of nitrate N and ammonium N in the soil treated with biostimulants were significantly lower than those in CK1 and CK2. Particularly, the soil nitrate and nitrogen contents decreased from 3.9% to 6.1% and 3.1% to 5.3% in the first year and from 3.3% to 5.6% and 3.8% to 6.1% in the second year. Meanwhile, in the first year, the ammonium N content also decreased significantly by 6.1% to 11.4% and 5.8% to 11.1%, respectively (Table 3).

**Table 3.** Effect of soil nitrate and ammonium N contents in different growth periods under the combined treatments with biostimulants.

| Content | Treatment | Wheat (2021–2022) | | | Wheat (2022–2023) | | | Maize (2021–2022) | | Maize (2022–2023) | |
|---|---|---|---|---|---|---|---|---|---|---|---|
| | | Jointing Stage | Flowering Stage | Maturation Stage | Jointing Stage | Flowering Stage | Maturation Stage | Flowering Stage | Maturation Stage | Flowering Stage | Maturation Stage |
| $NO_3^-$-N $(mg \cdot kg^{-1})$ | Ck1 | 17.81 ± 0.79 a | 20.42 ± 0.34 a | 10.85 ± 0.86 a | 17.74 ± 0.12 a | 20.5 ± 0.18 ab | 11.85 ± 0.2 a | 30.69 ± 0.57 a | 13.12 ± 0.44 a | 28.52 ± 0.43 c | 12.56 ± 0.23 a |
| | Ck2 | 17.75 ± 0.40 a | 20.39 ± 0.68 a | 10.53 ± 0.45 a | 17.87 ± 0.2 a | 20.67 ± 0.35 a | 11.76 ± 0.33 a | 30.6 ± 0.61 a | 13.01 ± 0.21 a | 28.76 ± 0.28 bc | 12.63 ± 0.21 a |
| | jf-20% | 17.23 ± 0.13 ab | 18.82 ± 0.66 bc | 9.00 ± 0.74 bc | 16.58 ± 0.29 bc | 19.96 ± 0.32 bc | 10.67 ± 0.18 b | 30.45 ± 0.43 a | 12.61 ± 0.13 b | 29.25 ± 0.23 a | 12.03 ± 0.09 b |
| | jf-30% | 17.32 ± 0.48 ab | 18.11 ± 0.49 c | 8.48 ± 0.24 c | 16.81 ± 0.18 bc | 19.54 ± 0.14 c | 10.52 ± 0.2 b | 30.45 ± 0.32 a | 12.46 ± 0.22 b | 29.12 ± 0.35 a | 11.92 ± 0.13 b |
| | jn-20% | 17.74 ± 0.67 a | 19.01 ± 0.23 b | 9.23 ± 0.54 b | 16.96 ± 0.29 b | 20.37 ± 0.35 ab | 10.15 ± 0.15 bc | 30.41 ± 0.56 ab | 12.51 ± 0.25 b | 29.16 ± 0.13 a | 12.05 ± 0.11 b |
| | jn-30% | 17.06 ± 0.32 ab | 18.42 ± 1.23 bc | 8.59 ± 0.33 bc | 16.65 ± 0.27 bc | 19.95 ± 0.23 bc | 9.87 ± 0.23 c | 29.53 ± 1.01 c | 12.32 ± 0.34 b | 29.12 ± 0.38 a | 12.15 ± 0.21 b |
| | jn-40% | 16.37 ± 0.84 b | 18.16 ± 0.17 c | 8.31 ± 0.30 c | 16.55 ± 0.37 c | 19.61 ± 0.17 c | 9.64 ± 0.47 c | 29.8 ± 0.53 bc | 12.34 ± 0.32 b | 29.02 ± 0.17 ab | 11.86 ± 0.17 b |
| | jny-27% | 17.62 ± 0.59 ab | 18.90 ± 0.26 bc | 9.24 ± 0.30 b | 16.96 ± 0.31 b | 20.44 ± 0.33 ab | 10.52 ± 0.19 b | 30.45 ± 0.49 a | 12.53 ± 0.27 b | 29.21 ± 0.14 a | 12.11 ± 0.13 b |
| $NH_4^+$-N $(mg \cdot kg^{-1})$ | Ck1 | 5.95 ± 0.16 a | 8.98 ± 0.18 a | 4.90 ± 0.54 a | 5.93 ± 0.21 a | 8.88 ± 0.41 ab | 5.29 ± 0.26 a | 13.89 ± 0.26 a | 7.97 ± 0.28 a | 12.93 ± 0.21 bc | 7.41 ± 0.13 ab |
| | Ck2 | 5.97 ± 0.21 a | 9.07 ± 0.28 a | 4.90 ± 0.18 a | 5.93 ± 0.12 a | 8.96 ± 0.3 a | 5.25 ± 0.21 a | 13.81 ± 0.36 a | 7.95 ± 0.24 a | 12.87 ± 0.30 c | 7.42 ± 0.25 ab |
| | jf-20% | 5.66 ± 0.11 b | 7.94 ± 0.15 bcd | 3.94 ± 0.19 b | 5.69 ± 0.17 b | 8.83 ± 0.23 ab | 5.1 ± 0.18 b | 13.72 ± 0.25 a | 7.35 ± 0.24 bc | 13.11 ± 0.11 abc | 7.30 ± 0.29 b |
| | jf-30% | 5.32 ± 0.17 c | 7.86 ± 0.15 cd | 3.99 ± 0.13 b | 5.58 ± 0.16 cd | 8.73 ± 0.17 ab | 5.08 ± 0.12 bc | 13.53 ± 0.40 a | 7.23 ± 0.12 bc | 12.96 ± 0.22 bc | 7.44 ± 0.31 a |
| | jn-20% | 5.89 ± 0.17 ab | 8.16 ± 0.14 bc | 4.11 ± 0.12 b | 5.65 ± 0.25 bc | 8.72 ± 0.11 ab | 5.15 ± 0.16 b | 13.44 ± 0.13 a | 7.2 ± 0.28 bc | 13.04 ± 0.15 abc | 7.35 ± 0.10 ab |
| | jn-30% | 5.35 ± 0.27 c | 8.20 ± 0.19 b | 3.92 ± 0.13 b | 5.48 ± 0.33 d | 8.55 ± 0.17 bc | 5.10 ± 0.24 b | 12.77 ± 0.32 b | 7.23 ± 0.25 bc | 13.3 ± 0.11 a | 7.36 ± 0.33 ab |
| | jn-40% | 5.15 ± 0.12 c | 7.68 ± 0.18 d | 3.99 ± 0.09 b | 5.50 ± 0.18 d | 8.26 ± 0.10 c | 5.00 ± 0.21 c | 12.73 ± 0.39 b | 7.06 ± 0.15 c | 13.19 ± 0.06 ab | 7.33 ± 0.12 ab |
| | jny-27% | 5.78 ± 0.15 ab | 8.09 ± 0.22 bc | 4.09 ± 0.18 b | 5.66 ± 0.22 bc | 8.66 ± 0.39 ab | 5.14 ± 0.23 b | 13.50 ± 0.41 a | 7.49 ± 0.2 b | 13.09 ± 0.16 abc | 7.36 ± 0.16 ab |

Values in the same column followed by different letters indicate significant differences ($p < 0.05$); $n = 3$.

### 3.2. Soil Enzyme Activity

Under the combined treatments with biostimulants, the soil urease activity fluctuated as wheat growth progressed. At first, it increased, then it decreased, and, finally, it increased. Similarly, for maize, the soil urease activity showed an increasing trend followed by a decreasing trend as growth progressed (Figure 2). At the jointing stage of wheat, the combined treatments with biostimulants had much lower soil urease activity than CK1 and CK2 (Figure 2a,d). Conversely, at the flowering stage, the urease activity increased. In 2021–2022, the urease activity increased significantly by 2.0% to 8.7% and 2.8% to 9.5% (Figure 2b), and in 2022–2023, it increased by 3.4% to 7.7% and 3.4% to 7.7% (Figure 2e). Changes in soil urease activity caused by biostimulants were the same at the maturation stage as they were at the jointing stage—both were below CK1 and CK2 levels (Figure 2c,f). In addition, as for maize, when combined with biostimulant, the urease activity increased at the flowering stage compared to CK1 and CK2. In 2021–2022, the increases were 1.9% to 10.7% and 2.5% to 11.3%, respectively (Figure 2g). In 2022–2023, the increases were 3.1% to 10.4% and 0.8% to 7.9%, respectively (Figure 2i). However, at maturity, the urease activities were below CK1 and CK2 levels (Figure 2h,j).

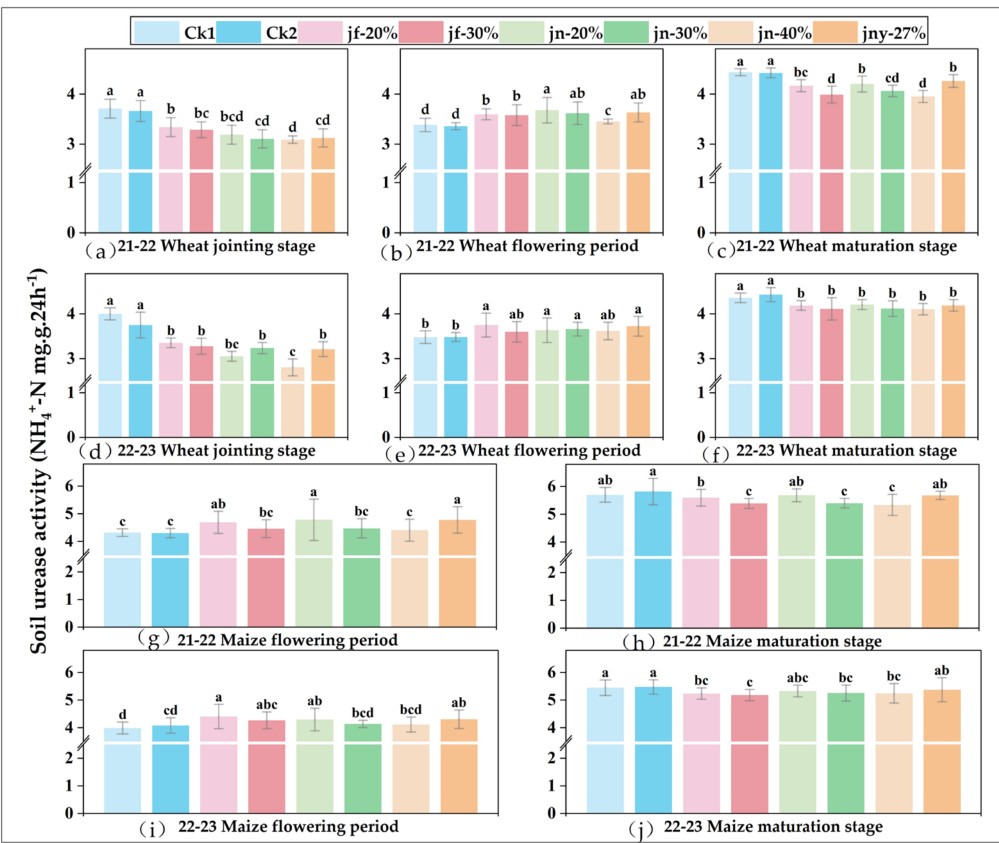

**Figure 2.** Soil urease activity in different growth periods under the combined treatments with biostimulants. Soil urease activity of wheat during the jointing stage (**a,d**), flowering stage (**b,e**), and maturation stage (**c,f**) in 2021–2022 and 2022–2023. Soil urease activity of maize during the flowering stage (**g,i**) and maturation stage (**h,j**) in 2021–2022 and 2022–2023. Different letters indicate significant differences ($p < 0.05$); 21–22 means 2021 to 2022; 22–23 means 2022 to 2023; $n = 3$.

The application of biostimulants combined with fertilizers decreased soil nitrate reductase activity during the jointing stage of wheat (Figure 3a,d). However, at the flowering stage, it showed varying degrees of increase compared to CK1 and CK2. In the first year, there was an increase of 0.3% to 3.6% and 0.7% to 3.8% (Figure 3b), and in the second year, there was an increase of 0.5% to 6.5% and 1.4% to 4.3% (Figure 3e). At the maturation stage,

soil nitrate reductase activity decreased below CK1 and CK2 (Figure 3c,f). For maize, after biostimulant treatment, the soil nitrate reductase activity in the flowering stage increased compared to CK1 and CK2. In the first year, there was an increase of 0.2% to 7.0% and 0.7% to 6.7% (Figure 3g), and in the second year, there was an increase of 0.1% to 3.1% and 0.9% to 4.5% (Figure 3i). However, at the maturation stage, the soil nitrate reductase activity decreased below CK1 and CK2 (Figure 3h,j).

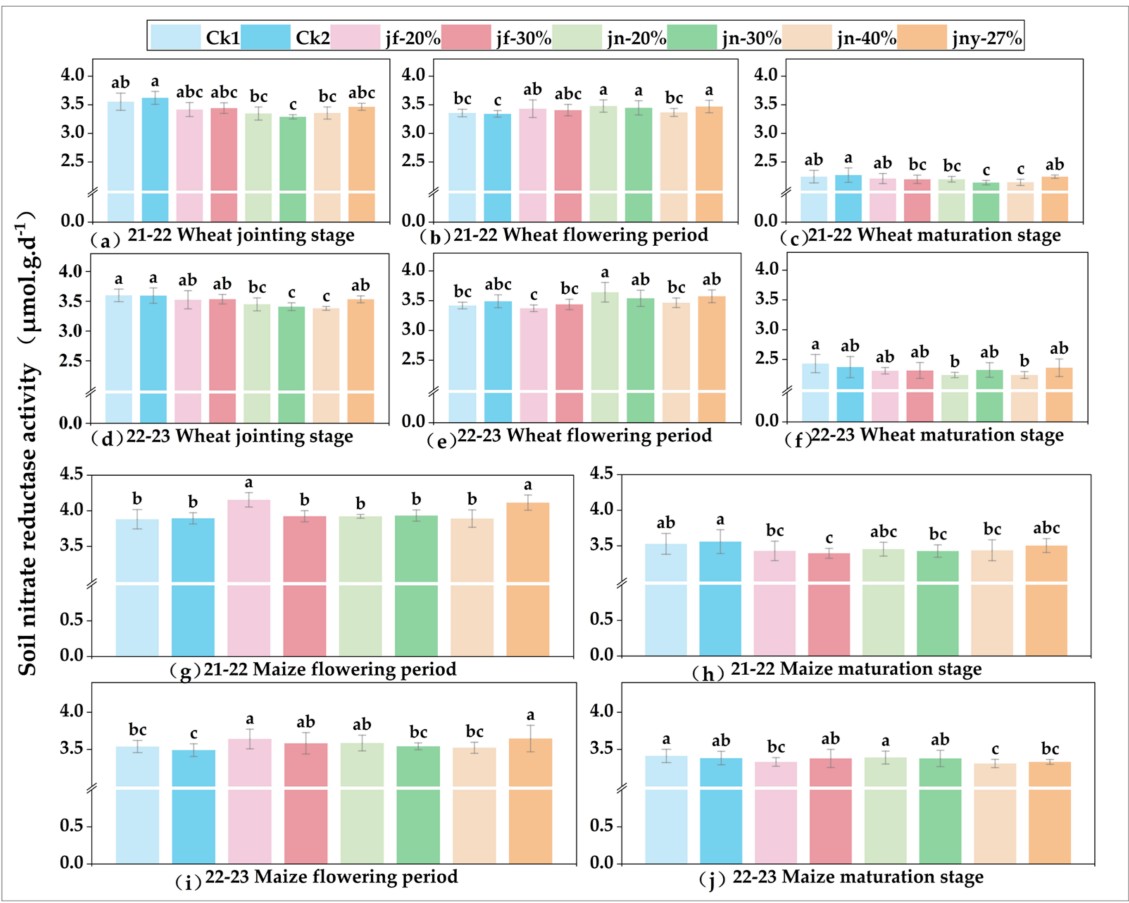

**Figure 3.** Soil nitrate reductase activity in different growth periods under the combined treatments with biostimulants. Soil nitrate reductase activity of wheat during the jointing stage (**a**,**d**), flowering stage (**b**,**e**), and maturation stage (**c**,**f**) in 2021–2022 and 2022–2023. Soil nitrate reductase activity of maize during the flowering stage (**g**,**i**) and maturation stage (**h**,**j**) in 2021–2022 and 2022–2023. Different letters indicate significant differences ($p < 0.05$); 21–22 means 2021 to 2022; 22–23 means 2022 to 2023; $n = 3$.

Biostimulant treatments decreased soil nitrite reductase activity during the wheat jointing stage (Figure 4a,d). However, during the flowering stage, activity increased compared to CK1 and CK2 to varying extents. In 2021–2022, the activity increased by 1.2% to 2.8% (Figure 4b); in 2022–2023, it increased by 0.02% to 1.1% (Figure 4e). Nevertheless, there were no significant differences in activity between the biostimulant and CK treatments at maturity (Figure 4c,f). Similarly, the soil nitrite reductase activity trend during the maize flowering stage was comparable to that of wheat (Figure 4g,i), but the overall activity was higher in maize. However, the nitrite reductase activity was lower than that of CK1 and CK2 at maturity.

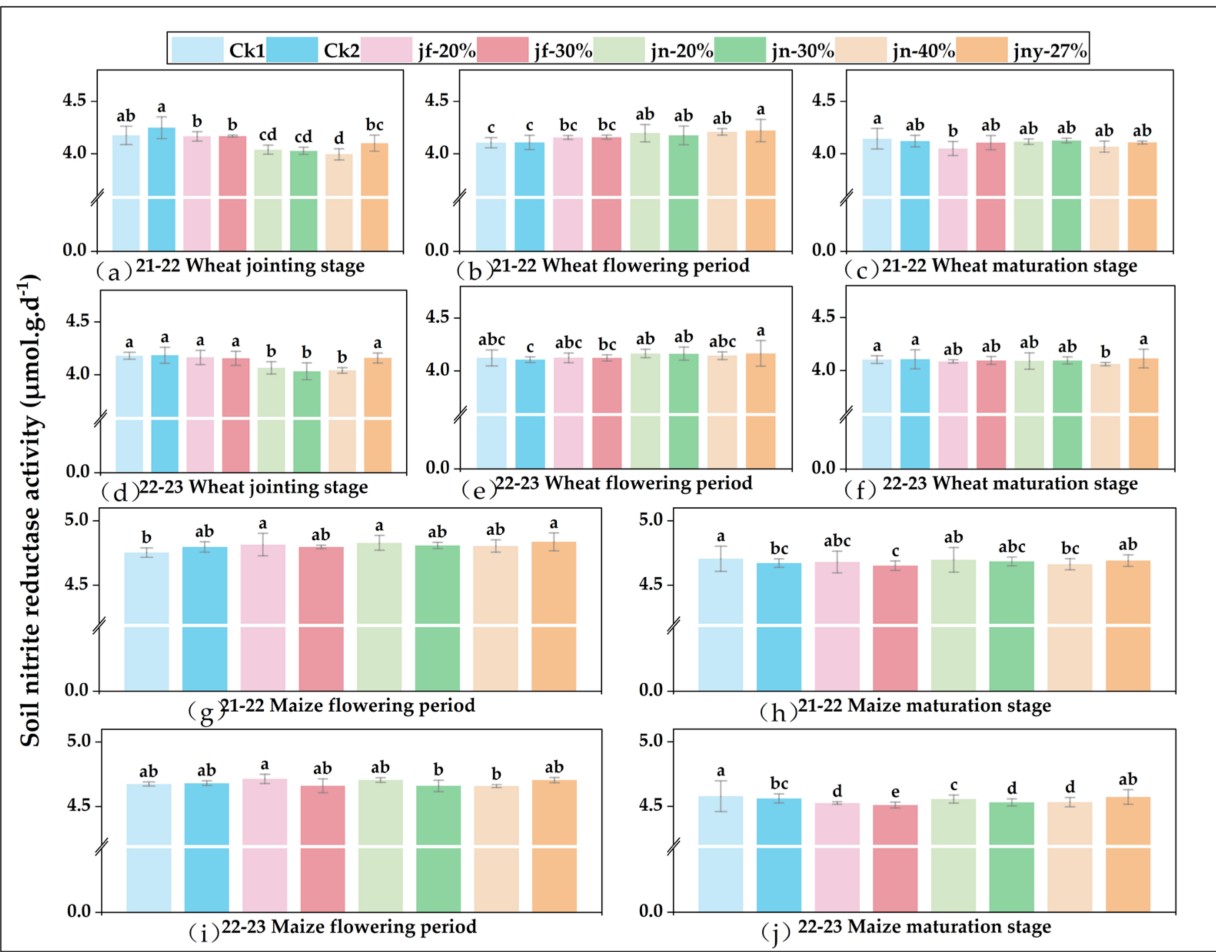

**Figure 4.** Soil nitrite reductase activity in different growth periods under the combined treatments with biostimulants. Soil nitrite reductase activity of wheat during the jointing stage (**a**,**d**), flowering stage (**b**,**e**), and maturation stage (**c**,**f**) in 2021–2022 and 2022–2023. Soil nitrite reductase activity of maize during the flowering stage (**g**,**i**) and maturation stage (**h**,**j**) in 2021–2022 and 2022–2023. Different letters indicate significant differences ($p < 0.05$); 21–22 means 2021 to 2022; 22–23 means 2022 to 2023; $n = 3$.

### 3.3. Plant N Transport and Accumulation

Biostimulant treatments enhanced N accumulation and transportation in wheat and maize (Table 4). In 2021–2022, N accumulation in wheat grain showed a significant increase of 4.0% to 8.0% and 2.5% to 6.5% in jn-20% and jny-27% treatments compared to CK1 and CK2, respectively. Similarly, the total N accumulation was significantly increased by 4.3% to 7.0% and 2.7% to 5.4% in the jn-20% and jny-27% treatments, respectively. However, all treatments exhibited lower pre-flowering N translocation compared to the CK1 and CK2 treatments, while post-flowering N assimilation showed a significant increase of 33.7% to 72.9% and 30.0% to 68.1% in 2021–2022 and 47.7% to 83.9% and 58.3% to 97.0% in 2022–2023, respectively. This trend was more pronounced in the jn treatments compared to the jny treatments. Further analysis revealed that biostimulant treatments increased post-flowering N assimilation's contribution to wheat grain N accumulation. Specifically, in 2021–2022, this contribution was significantly increased by 11.7% to 14.3% and 12.7% to 13.9% compared to CK1 and CK2 treatments, respectively (Table 4).

**Table 4.** Effect of N accumulation and translocation in different growth periods under the combined treatments with biostimulants (kg ha$^{-1}$).

| Content | Treatment | N Accumulation | | | | N Translocation (Pre-Flowering) | N Assimilation (Post-Flowering) | N Transfer Rate (Pre-Flowering %) | CGNT (%) | CGNA (%) |
|---|---|---|---|---|---|---|---|---|---|---|
| | | Vegetative Organs (Flowering Stage) | Vegetative Organs (Maturation Stage) | Grain (Maturation Stage) | Total Plant (Maturation Stage) | | | | | |
| Wheat (2021–2022) | CK0 | — | — | — | 91.3 | — | — | — | — | — |
| | Ck1 | 134.6 ± 6.3 a | 23.1 ± 1.2 b | 146.1 ± 2.3 cd | 169.3 ± 4.7 cd | 111.5 ± 4.3 a | 34.7 ± 2.3 e | 82.8 ± 1.1 a | 76.3 ± 1.6 a | 23.7 ± 1.2 b |
| | Ck2 | 136.1 ± 5.3 a | 23.7 ± 0.9 b | 148.2 ± 4.9 bc | 171.8 ± 6.1 c | 112.4 ± 5.1 a | 35.7 ± 2.3 e | 82.6 ± 1.3 a | 75.9 ± 1.6 a | 24.1 ± 1.3 b |
| | jf-20% | 115.2 ± 4.5 b | 21.8 ± 0.7 c | 147.9 ± 4.2 bc | 169.7 ± 5.6 cd | 93.4 ± 4.3 b | 54.5 ± 2.1 bc | 81.1 ± 1.3 b | 63.2 ± 1.4 b | 36.8 ± 1.0 a |
| | jf-30% | 106.1 ± 3.5 c | 21.5 ± 0.7 c | 133.1 ± 6.2 e | 154.6 ± 4.8 e | 84.5 ± 3.5 cd | 48.6 ± 3.5 d | 79.7 ± 1.7 bc | 63.5 ± 2.6 b | 36.5 ± 1.4 a |
| | jn-20% | 118.8 ± 4.3 b | 24.6 ± 0.5 a | 151.9 ± 4.9 b | 176.5 ± 3.8 b | 94.2 ± 4.3 b | 57.7 ± 4.3 ab | 79.3 ± 1.8 c | 62.0 ± 2.8 b | 38 ± 2.1 a |
| | jn-30% | 115.2 ± 4.7 b | 23.0 ± 0.3 b | 142.7 ± 6.7 d | 165.7 ± 3.4 d | 92.2 ± 4.7 bc | 50.5 ± 2.2 cd | 80.0 ± 0.8 bc | 64.6 ± 0.5 b | 35.4 ± 0.4 a |
| | jn-40% | 104.0 ± 3.7 c | 21.4 ± 0.3 c | 128.9 ± 7.5 e | 150.4 ± 6.7 e | 82.6 ± 5.2 d | 46.4 ± 2.6 d | 79.4 ± 0.9 c | 64.1 ± 0.5 b | 35.9 ± 0.7 a |
| | jny-27% | 121.1 ± 7.1 b | 23.3 ± 0.2 b | 157.8 ± 6.5 a | 181.1 ± 6.5 a | 97.8 ± 7.1 b | 60.0 ± 7.3 a | 80.7 ± 1.1 bc | 62.0 ± 4.5 b | 38 ± 1.2 a |
| Wheat (2022–2023) | CK0 | — | — | — | 90.7 | — | — | — | — | — |
| | Ck1 | 136.4 ± 3.2 a | 26.3 ± 1.2 a | 147 ± 5.5 abc | 173.3 ± 5.4 ab | 110.2 ± 3.5 a | 36.9 ± 3.7 c | 80.7 ± 1.4 ab | 75 ± 2.7 a | 25 ± 1 d |
| | Ck2 | 138.6 ± 2.5 a | 26.3 ± 1.2 a | 146.8 ± 1.2 bc | 173.0 ± 2 ab | 112.3 ± 4.1 a | 34.4 ± 2.4 c | 81.0 ± 1.4 a | 76.5 ± 0.9 a | 23.5 ± 1.1 d |
| | jf-20% | 118.1 ± 1.6 b | 22.8 ± 1.8 b | 149.8 ± 1.3 abc | 172.6 ± 2.9 ab | 95.3 ± 2.6 b | 54.5 ± 4.5 b | 80.7 ± 1.3 ab | 63.6 ± 1.3 b | 36.4 ± 1.5 c |
| | jf-30% | 109.8 ± 2.8 cd | 22.1 ± 1.5 b | 143.2 ± 5.1 c | 165.3 ± 6.6 c | 87.7 ± 2.3 c | 55.5 ± 5.2 b | 79.9 ± 1.5 ab | 61.4 ± 3.8 bc | 38.6 ± 1.4 bc |
| | jn-20% | 108.2 ± 6.2 cde | 22.6 ± 2.9 b | 153.4 ± 1.7 ab | 176.0 ± 4.4 a | 85.5 ± 3.4 cd | 67.8 ± 2.4 a | 79.1 ± 1.5 b | 55.8 ± 1.8 d | 44.2 ± 1.7 a |
| | jn-30% | 105.4 ± 2.1 de | 22.2 ± 1.7 b | 145.7 ± 7.4 c | 167.9 ± 7.1 bc | 83.2 ± 2.7 d | 62.5 ± 6.2 ab | 79.0 ± 1.8 b | 57.2 ± 3.1 cd | 42.8 ± 1.2 ab |
| | jn-40% | 103.3 ± 5.8 e | 21.3 ± 2.1 b | 143.6 ± 3.2 c | 164.9 ± 1.9 c | 82.1 ± 3.8 d | 61.6 ± 6.5 ab | 79.4 ± 0.9 ab | 57.2 ± 3.6 cd | 42.8 ± 1.4 ab |
| | jny-27% | 111.9 ± 2.9 c | 23 ± 0.8 b | 153.7 ± 1.5 a | 176.7 ± 2.2 a | 88.9 ± 2.8 c | 64.8 ± 3.7 a | 79.4 ± 0.7 ab | 57.8 ± 2.2 cd | 42.2 ± 1.2 ab |
| Maize (2021–2022) | CK0 | — | — | — | 100.53 | — | — | — | — | — |
| | Ck1 | 100.7 ± 2.4 e | 50.3 ± 1.6 c | 124.2 ± 3.6 d | 174.6 ± 5.9 d | 50.4 ± 1.5 d | 73.9 ± 2.4 cd | 50.0 ± 0.8 c | 40.6 ± 0.4 cd | 59.4 ± 0.5 bc |
| | Ck2 | 101.4 ± 3.8 de | 50.4 ± 1.7 c | 123.8 ± 3.1 d | 174.2 ± 6.9 d | 50.9 ± 1.1 cd | 72.8 ± 2.2 d | 50.3 ± 0.9 c | 41.2 ± 0.4 bc | 58.8 ± 0.4 cd |
| | jf-20% | 105.8 ± 3.4 b | 50.7 ± 1.7 bc | 130.0 ± 4.4 c | 180.7 ± 7.2 c | 55.1 ± 2.5 a | 74.9 ± 1.9 cd | 52.1 ± 1.1 a | 42.4 ± 0.8 a | 57.6 ± 0.7 e |
| | jf-30% | 101.7 ± 4.2 d | 48.9 ± 0.8 d | 125.5 ± 2.8 d | 174.4 ± 6.9 d | 52.7 ± 2.1 b | 72.8 ± 1.8 d | 51.9 ± 1.3 a | 42.0 ± 0.2 ab | 58.0 ± 0.3 de |
| | jn-20% | 108.6 ± 2.0 a | 52.3 ± 1.1 a | 141.0 ± 3.5 a | 193.3 ± 5.9 a | 56.3 ± 2.3 a | 84.7 ± 2.9 a | 51.8 ± 1.1 a | 39.9 ± 1 cde | 60.1 ± 0.9 abc |
| | jn-30% | 103.8 ± 3.1 c | 51.6 ± 1.7 ab | 132.7 ± 3.8 b | 184.4 ± 5.2 b | 52.1 ± 0.9 bc | 80.6 ± 2.1 b | 50.2 ± 0.6 c | 39.3 ± 0.8 e | 60.7 ± 0.4 a |
| | jn-40% | 98.4 ± 2.1 f | 48.6 ± 1.2 d | 126.1 ± 1.9 d | 174.7 ± 3.4 d | 49.9 ± 1.0 d | 76.2 ± 1.4 c | 50.6 ± 1 bc | 39.5 ± 1 de | 60.5 ± 0.5 ab |
| | jny-27% | 108.5 ± 4.3 a | 52.5 ± 1.9 a | 139.0 ± 3.3 a | 191.5 ± 7.4 a | 56 ± 1.9 a | 83.1 ± 3.1 ab | 51.6 ± 0.7 ab | 40.3 ± 0.1 cde | 59.7 ± 0.8 abc |

**Table 4.** *Cont.*

| Content | Treatment | N Accumulation | | | | N Transloca-tion (Pre-Flowering) | N Assimi-lation (Post-Flowering) | N Transfer Rate (Pre-Flowering %) | CGNT (%) | CGNA (%) |
|---|---|---|---|---|---|---|---|---|---|---|
| | | Vegetative Organs (Flowering Stage) | Vegetative Organs (Maturation Stage) | Grain (Maturation Stage) | Total Plant (Maturation Stage) | | | | | |
| Maize (2022–20233) | CK0 | — | — | — | 77.2 | — | — | — | — | — |
| | Ck1 | 84.3 ± 1.6 a | 40.8 ± 0.5 ab | 111.5 ± 4.4 cd | 153.9 ± 3.5 b | 43.5 ± 1.8 b | 67.9 ± 1.9 d | 51.6 ± 1.3 ab | 39.1 ± 1.3 ab | 60.9 ± 1.3 cd |
| | Ck2 | 84.7 ± 1.5 a | 41.3 ± 1.2 a | 110.3 ± 2.7 cd | 152.6 ± 4.5 b | 43.5 ± 1.9 b | 66.8 ± 1.3 d | 51.3 ± 1.2 b | 39.4 ± 1.1 ab | 60.6 ± 1.6 cd |
| | jf-20% | 85.0 ± 1.8 a | 40.2 ± 1.1 bc | 118.0 ± 7.5 b | 160.4 ± 5.1 a | 44.8 ± 1.1 a | 73.2 ± 1.1 b | 52.7 ± 1.4 a | 38.0 ± 0.9 b | 62.0 ± 2.5 c |
| | jf-30% | 82.8 ± 1.7 b | 40.0 ± 1.3 bcd | 107.3 ± 3.2 cd | 148.4 ± 3.6 cd | 42.8 ± 1.3 b | 64.5 ± 2.2 d | 51.7 ± 1.1 ab | 39.9 ± 1.3 a | 60.1 ± 1.3 d |
| | jn-20% | 79.3 ± 1.3 c | 39.5 ± 0.9 cd | 124.1 ± 4.2 a | 163.6 ± 4.2 a | 39.7 ± 0.9 c | 84.4 ± 1.0 a | 50.1 ± 0.9 c | 32.0 ± 0.7 d | 68.0 ± 0.7 a |
| | jn-30% | 78.2 ± 1.2 d | 39.3 ± 0.8 d | 111.5 ± 3.7 c | 152.0 ± 2.7 bc | 38.9 ± 1.3 c | 72.6 ± 2.1 bc | 49.7 ± 0.9 c | 34.9 ± 0.7 c | 65.1 ± 0.7 b |
| | jn-40% | 78.2 ± 0.7 d | 39.5 ± 0.8 cd | 106.9 ± 3.3 d | 147.6 ± 3.1 d | 38.7 ± 1.0 c | 68.2 ± 1.3 cd | 49.5 ± 0.6 c | 36.3 ± 1.3 c | 63.7 ± 1.3 b |
| | jny-27% | 79.5 ± 1.4 c | 40.0 ± 1.0 bcd | 122.5 ± 3.9 ab | 163.4 ± 3.2 a | 39.5 ± 1.5 c | 83.0 ± 1.4 a | 49.7 ± 0.9 c | 32.3 ± 0.6 d | 67.7 ± 0.6 a |

Values within the same column followed by different letters indicate significant differences ($p < 0.05$). CGNT: contribution to grain N of pre-flowering N translocation; CGNA: contribution to grain N of post-flowering N assimilation. CK0 was only used to calculate related indexes of N utilization and did not participate in ANOVA; $n = 3$.

Biostimulant treatments contributed to the total N accumulation in the reproductive organs of summer maize at maturity. Compared to CK1 and CK2, the treatments jf-20%, jn-20%, and jny-27% showed a significant improvement of 4.6% to 13.5% and 5.0% to 13.9% in 2021–2022 and 5.9% to 11.3% and 7.0% to 12.5% in 2022–2023, respectively. Additionally, biostimulant treatments reduced pre-flowering N translocation and increased post-flowering N assimilation in summer maize. This effect was particularly observed in jn and jny treatments, which significantly increased post-flowering N assimilation in maize by 3.2% to 12.5% and 4.7% to 14.0% in 2021–2022 and 7.0% to 24.2% and 8.7% to 24.2% in 2022–2023. Furthermore, biostimulant treatments increased the contribution of post-flowering N assimilation to kernel N accumulation. The jn and jny treatments showed a significant increase of 2.8% to 7.1% and 3.2% to 7.4% in 2022–2023, compared to CK1 and CK2 (Table 4).

### 3.4. Yield and Yield Composition

The application of biostimulants within the appropriate N reduction was found to maintain the stability of wheat and maize yields (Table 5). From 2021 to 2022, the fertile spike numbers of wheat treated with biostimulants showed no significant difference compared to CK1 and CK2. However, apart from the two treatments of jf-30% and jn-40%, the other treatments increased by 1.1% to 5.9% compared to CK1. Specifically, jf-20%, jn-20%, and jny-27% showed higher spike numbers of 0.2% to 2.9%. In 2022–2023, under the treatments of jn-20%, jn-30%, and jny-27%, the fertile spikes of wheat increased significantly by 4.5% to 6.7% compared to CK2. Except for the reduction in jn-40% treatment, the fertile spike numbers in the other treatments showed no significant difference compared to CK1 and CK2. Additionally, there was no significant difference in yield after the application of biostimulants, except for the reduction of jf-30% and jn-40% (Table 5).

**Table 5.** Effect of yield and yield composition under the combined treatments with biostimulants.

| Year | Treatment | Wheat | | | | Maize | | | |
|---|---|---|---|---|---|---|---|---|---|
| | | Fertile Spike Numbers (10⁴ ha⁻¹) | Grain Numbers per Spike | 1000-Grain Weight (g) | Yield (kg ha⁻¹) | Row Numbers | Kernel Numbers | 500-Grain Weight (g) | Yield (kg ha⁻¹) |
| 2021–2022 | Ck0 | 331.0 | 29.8 | 50.0 | 4284.5 | 11.8 | 27.8 | 164.6 | 5431.3 |
| | Ck1 | 413.0 ± 9.3 ab | 48.5 ± 1.2 a | 50.7 ± 0.8 ab | 7618.7 ± 82.5 abc | 15.5 ± 0.1 b | 39.8 ± 1.1 b | 166.9 ± 0.6 a | 9766.8 ± 121 cd |
| | Ck2 | 422.7 ± 10.2 ab | 48.7 ± 0.9 a | 49.0 ± 0.5 c | 7782.8 ± 49.9 ab | 15.6 ± 0.1 b | 39.7 ± 0.9 b | 167.2 ± 1.1 a | 9738.2 ± 136.7 d |
| | jf-20% | 417.7 ± 9.3 ab | 46.5 ± 1.1 b | 49.1 ± 1.1 c | 7633 ± 68.7 abc | 15.8 ± 0.2 ab | 40.9 ± 1.1 a | 166.9 ± 0.5 a | 9957.4 ± 43.7 b |
| | jf-30% | 402.0 ± 8.3 b | 46.0 ± 1.2 b | 48.9 ± 0.9 c | 7533.5 ± 36.6 c | 15.9 ± 0.1 ab | 40.2 ± 1.3 ab | 166.6 ± 1.3 a | 9804.9 ± 124.6 c |
| | jn-20% | 425.3 ± 8.2 ab | 49.2 ± 1.4 a | 49.5 ± 0.9 bc | 7820.6 ± 101.4 a | 15.9 ± 0.2 ab | 40.8 ± 0.8 a | 167.7 ± 0.4 a | 10214.6 ± 127.4 a |
| | jn-30% | 419.7 ± 7.3 ab | 46.3 ± 1.2 b | 50.7 ± 0.7 ab | 7562.5 ± 33.7 bc | 15.7 ± 0.1 ab | 39.5 ± 1.3 b | 167.1 ± 1.1 a | 9966.9 ± 46.5 b |
| | jn-40% | 401.0 ± 5.9 b | 44.8 ± 1.0 c | 51.5 ± 0.6 a | 7468.8 ± 53.9 c | 15.9 ± 0.1 ab | 39.7 ± 1.2 b | 167.3 ± 0.7 a | 9776.3 ± 57.2 cd |
| | jny-27% | 437.3 ± 7.8 a | 48.7 ± 1.5 a | 48.7 ± 1.2 c | 7840.4 ± 94.2 a | 16.2 ± 0.2 a | 40.8 ± 0.9 a | 167.8 ± 0.8 a | 10252.7 ± 131.8 a |
| 2022–2023 | Ck0 | 273 | 32.1 | 48.9 | 3528.4 | 12.3 | 20.1 | 129.6 | 3107.7 |
| | Ck1 | 389.0 ± 6.9 bcd | 48.5 ± 1.3 a | 50.0 ± 0.4 a | 7728.3 ± 45.9 ab | 15.2 ± 0.2 ab | 35 ± 1.3 ab | 134.3 ± 0.7 ab | 6680.5 ± 89.7 b |
| | Ck2 | 383.7 ± 4.7 cd | 48.3 ± 0.9 ab | 49.6 ± 0.2 a | 7608.6 ± 79.4 c | 15.2 ± 0.1 ab | 34.5 ± 1.8 ab | 134.7 ± 0.6 ab | 6516.4 ± 118.4 bc |
| | jf-20% | 397.7 ± 2.9 abc | 48.1 ± 1.3 ab | 49.7 ± 0.4 a | 7634.5 ± 20.2 bc | 15.4 ± 0.3 ab | 35.5 ± 1.4 ab | 134.8 ± 0.6 a | 6950.8 ± 94.4 a |
| | jf-30% | 373.7 ± 8.8 d | 48.0 ± 1.2 b | 49.5 ± 0.3 a | 7443.6 ± 51.4 d | 15.0 ± 0.1 ab | 34.9 ± 0.6 ab | 132.7 ± 0.4 ab | 6643.7 ± 49.5 b |
| | jn-20% | 408.3 ± 3.9 a | 48.2 ± 0.7 ab | 49.8 ± 0.8 a | 7725.1 ± 88.9 ab | 15.7 ± 0.1 a | 36.3 ± 1.0 a | 134 ± 0.7 ab | 6986.4 ± 40 a |
| | jn-30% | 401.0 ± 10.5 ab | 48.1 ± 0.6 ab | 49.5 ± 0.7 a | 7653.9 ± 36.7 bc | 15.6 ± 0.3 ab | 33.7 ± 1.2 bc | 132.1 ± 0.3 b | 6566.4 ± 86.3 bc |
| | jn-40% | 380.0 ± 7.2 d | 47.9 ± 0.5 b | 49.7 ± 0.7 a | 7440.4 ± 112.5 d | 15.3 ± 0.6 ab | 32.7 ± 1.3 c | 132.8 ± 0.8 ab | 6369.5 ± 126.6 c |
| | jny-27% | 409.3 ± 9.3 a | 48.3 ± 1.1 ab | 50.0 ± 0.1 a | 7812.4 ± 54.9 a | 15.7 ± 0.1 a | 35.8 ± 1.1 a | 134.8 ± 0.8 a | 6891.1 ± 116.7 a |

Values within the same column followed by different letters indicate significant differences ($p < 0.05$). CK0 was only used to calculate related indexes of N utilization and did not participate in ANOVA; $n = 3$.

From 2021 to 2022, the row numbers of maize under biostimulant treatments increased by 1.3% to 4.5% and 0.6% to 3.9% compared to CK1 and CK2, respectively. The most significant increase was observed in the jny-27% treatment. The kernel numbers significantly increased by 2.5% to 2.8% and 2.8% to 3.0% under the jf-20%, jn-20%, and jny-27% treatments compared to CK1 and CK2, respectively. Regarding yield, the biostimulant treatments showed a significant increase of 2.0% to 5.0% and 2.3% to 5.3% compared to CK1 and CK2, respectively, except for jf-30% and jn-40%. In 2022–2023, kernel numbers showed no significant difference, except for jn-40%, which showed a decrease. Furthermore, the jf-20%, jn-20%, and jny-27% treatments exhibited significant yield improvements of 3.2% to 4.6% and 5.8% to 7.2% for CK1 and CK2, respectively (Table 5).

### 3.5. N Utilization

The combined application of biostimulants and fertilizers with N reduction significantly improved the NUE and N partial productivity (NPP) of wheat and maize, and the N harvest index (NHI) also increased (Table 6). Compared with CK1 and CK2, the NUE of biostimulant–treated wheat was significantly increased by 6.7–24.0% and 5.3–22.6% in 2021–2022, and that of maize was significantly increased by 11.6–22.6% and 11.8–22.8%, respectively. In addition, the NPP of wheat was significantly increased by 25.4%–63.6% and 22.7–60.0%, and that of maize was significantly increased by 27.4–66.8% and 27.8–67.3%

compared to CK1 and CK2, respectively. In 2022–2023, NUE under biostimulant treatment was significantly higher by 7.6% to 14.4% and 8.3% to 15.2% in wheat and 11.1% to 18.5% and 11.7% to 19.1% in maize, respectively, compared to CK1 and CK2. In addition, the NPP was significantly higher in wheat by 22.6% to 58.8% and 25.0% to 61.9% and in maize by 27.4% to 66.8% and 27.8% to 67.3% compared to CK1 and CK2, respectively (Table 6).

**Table 6.** Effect of N utilization under the combined treatments with biostimulants.

| Year | Treatment | Wheat | | | Maize | | |
|---|---|---|---|---|---|---|---|
| | | NUE (%) | NPP (kg·kg$^{-1}$) | NHI (%) | NUE (%) | NPP (kg·kg$^{-1}$) | NHI (%) |
| 2021–2022 | Ck1 | 41.6 ± 1.5 b | 40.6 ± 0.3 b | 86.3 ± 0.1 b | 32.9 ± 0.3 f | 43.4 ± 0.5 f | 71.2 ± 0.2 d |
| | Ck2 | 43 ± 0.5 b | 41.5 ± 0.4 b | 86.2 ± 0.3 b | 32.7 ± 0.4 f | 43.3 ± 0.1 f | 71.1 ± 0.1 d |
| | jf-20% | 52.3 ± 2.4 a | 50.9 ± 0.4 a | 87.1 ± 0.7 a | 44.5 ± 0.9 e | 55.3 ± 0.2 e | 72.0 ± 0.2 c |
| | jf-30% | 48.3 ± 1.4 a | 57.4 ± 0.7 a | 86.1 ± 0.3 b | 46.9 ± 0.4 d | 62.3 ± 0.2 c | 71.9 ± 0.4 c |
| | jn-20% | 56.8 ± 2.6 a | 52.1 ± 0.3 a | 86.1 ± 0.2 b | 51.5 ± 1.6 c | 56.7 ± 0.4 d | 72.9 ± 0.4 a |
| | jn-30% | 56.7 ± 2.6 a | 57.6 ± 1.1 a | 86.1 ± 0.6 b | 53.2 ± 1.3 b | 63.3 ± 0.3 b | 72.0 ± 0.3 c |
| | jn-40% | 52.5 ± 0.6 a | 66.4 ± 1.0 a | 85.8 ± 0.2 b | 54.9 ± 0.9 a | 72.4 ± 1.1 a | 72.2 ± 0.3 bc |
| | jny-27% | 65.6 ± 1.0 a | 57.3 ± 0.3 a | 87.1 ± 0.2 a | 55.5 ± 1.4 a | 62.5 ± 0.7 c | 72.6 ± 0.6 ab |
| 2022–2023 | Ck1 | 44.7 ± 0.6 d | 42.1 ± 0.2 f | 84.8 ± 0.5 b | 34.1 ± 0.8 d | 30.8 ± 0.4 g | 72.4 ± 0.5 cd |
| | Ck2 | 43.9 ± 0.9 d | 41.3 ± 0.1 g | 84.8 ± 0.5 b | 33.5 ± 1.1 d | 30.0 ± 0.1 h | 72.3 ± 0.8 cd |
| | jf-20% | 52.3 ± 0.5 c | 51.7 ± 0.1 e | 86.8 ± 1.0 a | 46.3 ± 1.3 bc | 39.6 ± 0.5 f | 73.6 ± 1.1 c |
| | jf-30% | 53.1 ± 1.1 c | 57.7 ± 0.3 c | 86.6 ± 0.4 a | 45.2 ± 2.8 c | 42.7 ± 0.3 d | 72.4 ± 0.4 d |
| | jn-20% | 54.0 ± 0.8 c | 52.7 ± 0.2 d | 87.2 ± 0.4 a | 48.0 ± 1.3 b | 40.5 ± 0.2 e | 75.8 ± 0.8 a |
| | jn-30% | 54.3 ± 1.2 bc | 59.3 ± 0.2 b | 86.8 ± 0.5 a | 47.5 ± 0.6 b | 43.2 ± 0.2 c | 73.4 ± 0.7 b |
| | jn-40% | 59.1 ± 0.9 a | 66.9 ± 0.2 a | 87.1 ± 0.8 a | 52.2 ± 1.8 a | 49.1 ± 0.4 a | 72.4 ± 0.4 b |
| | jny-27% | 58.3 ± 1.1 ab | 57.8 ± 0.1 c | 87.0 ± 0.3 a | 52.6 ± 0.5 a | 44.1 ± 0.3 b | 75.0 ± 0.9 a |

Values within the same column followed by different letters indicate significant differences ($p < 0.05$). NUE: N use efficiency; NPP: N partial productivity; NHI: N harvest index; $n = 3$.

## 4. Discussion

### 4.1. Co-Application of Biostimulant and Fertilizer as Base Fertilizer in Soil Improves Nutrient Utilization

Seaweed polysaccharides and chitosan are biostimulants that regulate soil nutrient availability [33]. When seaweed polysaccharides are applied to the soil, they can retain soil nutrients through colloidal adsorption. This leads to increased nutrient availability and utilization during the mid-to-late stages of crop growth [21,34,35]. Similarly, chitosan exhibits good adsorption and slow-release properties. Its amino groups can form complexes with soil nutrients, resulting in slow nutrient release and improved nutrient availability [36,37]. The molecular chain structure of urea-formaldehyde, used in the jn treatment, helps reduce the rate of N release in the fertilizer. Similarly, jny treatment with organic fertilizer can enhance soil's physical and chemical properties, affecting nutrient absorption and utilization efficiency [38,39]. This study found that wheat and maize significantly improved N use efficiency and N partial productivity under the jf, jn, and jny fertilization treatments compared to common fertilization methods. Furthermore, the N use efficiency was higher in the jn and jny treatments compared to the jf treatment, which could be attributed to the combined effect of urea-formaldehyde and organic fertilizer.

### 4.2. Biostimulants Improve Nutrient Availability and Reduce Residues by Regulating Soil Enzyme Activity

Biostimulants have been found to significantly affect enzyme activity in soil. Studies on rape plants have shown that combining biostimulants with N reduction can increase the multiplication rate of soil microorganisms, which are typically present during the flowering stage. Consequently, the increase in soil enzyme activity from flowering to maturity will ultimately influence the transformation and availability of soil N [40]. Urease activity is crucial in converting urea to $NH_4^+$, making urease activity closely linked to soil

N levels [41]. Traditional fertilizers accelerate the hydrolytic conversion process of urea through urease, resulting in an early peak of ammonium N and high N loss [42]. However, in this study, after N reduction with a biostimulant, wheat and maize showed a decrease in soil urease activity during the early reproductive period and an increase during the mid-to-late reproductive period. This delayed the appearance of the ammonium N peak, which was more favorable for N accumulation in the mid-to-late stages of crop growth. Consequently, it provided a foundation for N conversion and improved N utilization. Soil nitrate reductase converts soil nitrate N into nitrite N, which is then catalyzed by nitrite reductase to form hydroxylamine ($NH_2OH$). This hydroxylamine is further catalyzed to use superscript and subscript for crop utilization [43]. In this study, applying biostimulants effectively increased soil nitrate reductase and nitrite reductase activities during the mid-to-late stages of wheat and maize. This led to better maintenance of nutrient levels in the soil and facilitated the accumulation of N above ground during the mid-to-late growth stages of crops. Soil ammonium and nitrate N contents are closely related to crop N accumulation. According to this study, the nitrate N and ammonium N amounts decreased in the early stages due to the effects of N reduction combined with biostimulants. However, the ammonium N and nitrate N peaks were observed in the mid-to-late stages. Soil ammonium N and nitrate N content also showed a tendency to decrease in the flowering stage of wheat and maize compared to conventional N application. This decrease can be attributed to soil nitrification and denitrification, as well as crop N uptake and utilization [44]. In the end, both ammonium and nitrate nitrogen residues in mature maize and wheat were much lower than they would have been with regular fertilization. This helped reduce environmental risks.

### 4.3. Biostimulants Combined with N Reduction Fertilizer Promoted Plant N Accumulation and Transportation

Both pre-flowering nutrient organ nitrogen transport and post-flowering assimilated nitrogen distribution have an impact on crop seed nitrogen accumulation. Improving the transport and distribution of N during both stages is an effective strategy to stabilize and increase crop yields [45]. Numerous studies have demonstrated the significant impact of biostimulants on enhancing N uptake and transportation in crops. For instance, utilizing biostimulants derived from seaweed extracts can effectively prolong the aging process of wheat flag leaves. Moreover, these biostimulants have been found to enhance N absorption during the late stages of reproduction, leading to a significant increase in N accumulation in wheat grains. Consequently, this promotes overall yield improvement in wheat crops [29]. In addition, applying seaweed extract at the 5th leaf stage and silking stage of maize can promote the accumulation of dry matter in the middle growth stage and, finally, increase the yield by 38% compared with control [28]. Under the influence of biostimulants, the N accumulation and transportation of wheat and maize plants during the mid-to-late stage of this experiment also exhibited alterations. The decreasing trend in crop N accumulation under jf, jn, and jny treatments at the flowering stage can be attributed to nutrient release regression. However, the crops maintained good growth in the mid-to-late stage. This enhanced post-flowering N uptake and translocation, leading to an increased contribution of post-flowering assimilation of N to seed N accumulation. Consequently, seed N accumulation was enhanced, ultimately promoting fertile spike numbers and grain numbers in wheat and increasing row numbers and kernel numbers in maize. Furthermore, jn and jny treatments had a more pronounced effect on promoting post-flowering N accumulation, which could be attributed to the combination of biostimulants and fertilizers.

### 4.4. Biostimulants Combined with N Reduction Fertilizer Increases Economic Efficiency

In addition, a well-designed fertilizer treatment needs to focus not only on fertilizer utilization and yield but also on ease of application and the ability to achieve higher economic efficiency. Biostimulants are commonly used as separate foliar sprays after

fertilizer application. Although this method can improve crop yield and N use efficiency, it increases labour costs [29,46,47]. In contrast, the one-time application of biostimulants combined with fertilizers as base fertilizer, as explored in this study, proved to be more convenient and efficient. In terms of cost, all biostimulant treatments decreased except for jn-20%, which showed a small increase in cost. The combination of appropriate fertilizer N reduction and biostimulants has been found to increase the economic benefits of wheat (Figure 5a,b) and maize (Figure 5c,d). This is primarily due to lower costs while maintaining a stable output. In addition, maize is more profitable than wheat. This can be attributed to the higher soil enzyme activity during the maize growing season, which results in greater soil transformation and nutrient supply intensity. As a result, maize has an increased degree of N transfer and distribution, leading to higher production and economic performance.

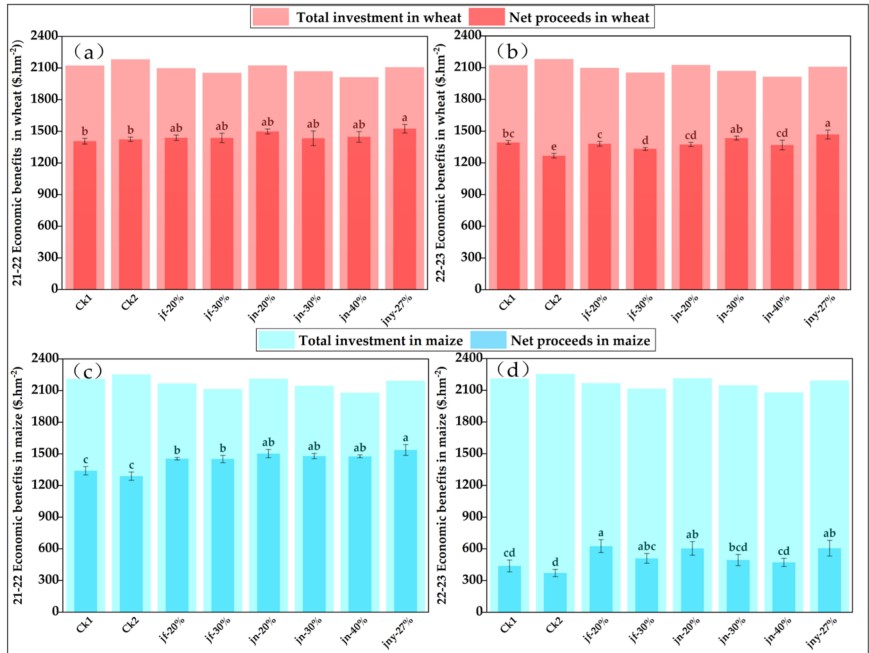

**Figure 5.** Effect of economic benefits of wheat and maize under the combined treatments with biostimulants. The total investment and net proceeds of wheat in 2021–2022 (**a**) and 2022–2023 (**b**); The total investment and net proceeds of maize in 2021–2022 (**c**) and 2022–2023 (**d**). Different letters indicate significant differences ($p < 0.05$); 21–22 means 2021 to 2022; 22–23 means 2022 to 2023. The costs are as follows: liquid biostimulant (21 USD per litre); common compound fertilizer (616 USD per ton); organic fertilizer (37.8 USD per ton); urea (588 USD per ton); biostimulant chelated urea-formaldehyde fertilizer (742 USD per ton); the total remaining field management fees, seed pesticide costs, and land rent (1470 USD per season); local wheat prices in 2021 and 2023 are 463.4 USD and 456.4 USD per ton, respectively; the prices of maize are 364 USD and 399 USD per ton, respectively; $n = 3$.

## 5. Conclusions

Biostimulants have been found to be beneficial in improving N use efficiency and yield of crops under low N applications. When a biostimulant is used in combination with a common compound fertilizer, a stable yield can be achieved with a 20% reduction in N dosage. Similarly, when a biostimulant is used with chelated urea-formaldehyde fertilizer, N reductions of 20–30% can be achieved while maintaining a stable yield. When a biostimulant is used with chelated urea-formaldehyde fertilizer and organic fertilizers, the yield goes up and the amount of nitrogen used goes down by 27%. When nitrogen reduction was used with a biostimulant, the peak of soil nutrient release was pushed back, and soil enzyme activity, nitrogen accumulation, and nitrogen transport were all improved during the middle to late stages of crop growth. Ultimately, this leads to improved N

utilization efficiency and economic benefits. Further research is required to investigate the nitrogen reduction effect of biostimulants combined with compound fertilizers in different soil textures and regions.

**Author Contributions:** Conceptualization, G.F.; data curation, J.L. and H.M. (Haiyan Ma); formal analysis, J.L. and H.M. (Haiyan Ma); funding acquisition, G.F.; investigation, H.M. (Hongliang Ma), X.H., D.H. and F.L.; methodology, J.L., H.M. (Haiyan Ma) and H.Y.; project administration, H.Y. and G.F.; supervision, G.F.; writing—original draft, J.L. and H.M. (Haiyan Ma); writing—review and editing, J.L., H.M. (Haiyan Ma), H.M. (Hongliang Ma), F.L. and X.H. All authors have read and agreed to the published version of the manuscript.

**Funding:** This research was funded by The Science and Technology Major Project of Sichuan Province (Grant No.: 2022ZDZX0014) and the Key Research and Development Project of Sichuan Province (Grant No.: 2021YFYZ0002).

**Data Availability Statement:** For additional information contact the author by correspondence.

**Conflicts of Interest:** The authors declare no conflict of interest.

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
