# Peer review of "Comprehensive Effects of N Reduction Combined with Biostimulants on N Use Efficiency and Yield of the Winter Wheat–Summer Maize Rotation System"

_agronomy, doi:10.3390/agronomy13092319_

Round 1
Reviewer 1 Report
Dear Authors
The manuscript ID agronomy-2595005, entitled "Comprehensive Effects of Nitrogen Reduction Combined with Biostimulant on Nitrogen use efficiency and Yield of Winter Wheat-Summer Maize Rotation System", presents an important research problem. The presented study shows the possibilities of reducing the level of plant fertilization with nitrogen, increasing the effectiveness of fertilization, improving profitability and reducing the dispersion of this macronutrient in the environment while maintaining the amount of yields obtained. The manuscript submitted for review should be published after minor corrections. Keywords should be changed as they were previously used in the title of the manuscript. In chapter 2 "Materials and Methods" you should: 1) provide the soil pH value; 2) add a comment on soil properties and meteorological data compared to the requirements of the tested plant species; 3) specify what organic fertilizer and what biostimulators were applied to the soil (brief characteristics), when they were applied (e.g. dates, ...); 4) a detailed description of agricultural technology should be completed; 5) add information on the performed statistical analysis of test results. In chapter 3 "Results", I suggest giving the percentage reduction or increase with accuracy to tenths of a percent.
With regards,
Reviewer
Author Response
Response to Reviewer 1 Comments
Dear reviewer 1:
Thank you for your comments concerning our manuscript entitled “Comprehensive Effects of Nitrogen Reduction Combined with Biostimulant on Nitrogen use efficiency and Yield of Winter Wheat-Summer Maize Rotation System (agronomy-2595005)”. These all comments are valuable and very helpful for revising and improving our MS, as well as are an important guidance to our researches. We have carefully checked and prepared the MS according to your suggestions. We hope that the revised manuscript will be according to your concerns. Revised portions are highlighted in yellow color of manuscript. The corrections in the paper and responses to the reviewer’s comments are as following:
Point 1: The presented study shows the possibilities of reducing the level of plant fertilization with nitrogen, increasing the effectiveness of fertilization, improving profitability and reducing the dispersion of this macronutrient in the environment while maintaining the amount of yields obtained.
Response 1: Thank you for your careful review and appreciations.
Point 2: Keywords should be changed as they were previously used in the title of the manuscript.
Response 2: Dear reviewer, thank you for this comment. We have tried our best to avoid repetition of keywords and title.
Title: Comprehensive Effects of N Reduction Combined with Biostimulant on N Use Efficiency and Yield of the Winter Wheat-Summer Maize Rotation System
Old Keyword: Biostimulant; wheat; maize; yield; nitrogen use efficiency
New Keywords: Winter wheat - Summer maize; Biostimulant; Nitrogen accumulation; Yield; Nitrogen use efficiency; Soil enzyme activity
Point 3: In chapter 2 "Materials and Methods" you should: 1) provide the soil pH value;
Response 3: Dear reviewer, thank you for this comment. We have provide the soil pH value in “Table 1. Soil fertility of the experimental site of two years”.
Point 4: 2) add a comment on soil properties and meteorological data compared to the requirements of the tested plant species;
Response 4: Dear reviewer, thank you for this comment. We have add a comment on soil properties and meteorological data compared to the requirements of the tested plant species as follows:
The meteorological data for the crop growth period in each quarter have been separated by dotted lines in the Figure 1, and the time for measuring basic fertility is the interval between maize harvest and wheat sowing.
Point 5: 3) specify what organic fertilizer and what biostimulators were applied to the soil (brief characteristics), when they were applied (e.g. dates, ...);
Response 5: Dear reviewer, thank you for this comment. We have specified the spicies of organic fertilizer and biostimulator that applied to the soil and the application time.
The wheat variety used in the experiment was Shu Mai 133 and the maize variety was Zheng Hong No.6. Fertilizer materials used included: biostimulant (homogeneous liquid), biostimulant chelated urea formaldehyde fertilizer (solid fertilizer with N:P:K ratio of 25:9:16). The biostimulant used in this experiment was a complex formed by polymerizing seaweed polysaccharide and chitosan with a nonionic active agent, which was obtained from Jijian Bio-technology Co (Chengdu, China). The organic fertilizer had a nitrogen, phosphorus and potassium ratio of 0.98:1:0.5. Common compound fertilizer had an N:P:K ratio of 26:10:16. Urea with an effective nitrogen content of 46%, organic fertilizer, common compound fertilizer, and urea were purchased from the local agri-cultural market.
Point 6: 4) a detailed description of agricultural technology should be completed;
Response 6: Dear reviewer, thank you for this comment. We have added a detailed description of agricultural techniques.
The winter wheat-summer maize crop rotation model was implemented in this study, with the previous crop being wheat. Following the harvest, all straw was re-moved from the experimental plot to ensure that soil fertility was not compromised. The plot was then tilled and sorted. Each plot in the experiment had an area of 21 m2 (4.2 × 5 m), and there were 3 replicates per treatment, resulting in a total of 27 plots. Wheat was sown in rows and covered with soil after sowing. The row spacing of each plot was 20 cm, with a total of 21 rows. The initial seedling density was set at 2.0 × 106 plants ha-1. Sowing dates for wheat were 27 October 2021 and 24 October 2022, with harvest dates of 8 May 2022 and 4 May 2023, respectively. The topdressing dates for CK2 were November 24, 2022 and November 22, 2023, respectively. For maize, it was sown in acupoints with a row spacing of 70cm and a hole spacing of 25cm. The initial seedling density at the four-leaf stage was 5.72 × 104 plants ha-1. Maize was sown on 13 May 2021 and 19 May 2022, respectively, and harvested on 2 September 2021 and 11 September 2022, respec-tively. Except urea as topdressing, other fertilizers were used as base fertilizer after crop seeding. The field management practices for wheat and maize were consistent with those followed in local high-yield fields.
Point 7: 5) add information on the performed statistical analysis of test results.
Response 7: Dear reviewer, thank you for this comment. We have added the information on the performed statistical analysis of test results.
2.4. Date Analysis
The Excel2010 software was used to collate the experimental data, the IBM SPSS Statistics 22.0 software was used for variance analysis (ANOVA). The least significant difference (LSD) method was used for significance test (p < 0.05). The image visualiza-tion was performed by Origin 2021.
Point 8: In chapter 3 "Results", I suggest giving the percentage reduction or increase with accuracy to tenths of a percent.
Response 8: Dear reviewer, thank you for this comment. We have changed the percentage with accuracy to tenths of a percent in chapter 3 "Results".
We appreciate your warm work earnestly and hope that the corrections will meet with approval. Once again, thank you very much for your comments and suggestions.
Yours sincerely
Junji Li
Corresponding author: Gaoqiong Fan
Email: fangao20056@126.com

Reviewer 2 Report
Dear authors,
I appreciate your hard-work and research. However, I think you should take time and look carefully at the manuscript before submission. I have put some comments in the attached file. Please modify them accordingly.

The paper is well-written but there are some issues which must be fixed. The authors should be more careful.
Author Response
Response to Reviewer 2 Comments
Dear reviewer 2:
Thank you for your comments concerning our manuscript entitled “Comprehensive Effects of Nitrogen Reduction Combined with Biostimulant on Nitrogen use efficiency and Yield of Winter Wheat-Summer Maize Rotation System (agronomy-2595005)”. These all comments are valuable and very helpful for revising and improving our MS, as well as are an important guidance to our researches. We have carefully checked and prepared the MS according to your suggestions. We hope that the revised manuscript will be according to your concerns. Revised portions are highlighted in yellow color of manuscript. The corrections in the paper and responses to the reviewer’s comments are as following:
Point 1: I appreciate your hard-work and research. However, I think you should take time and look carefully at the manuscript before submission.
Response 1: Dear reviewer, thank you for this comment. We are sorry for the lack of readability of the manuscript due to language problems and some omissions. We have made complete revisions in the manuscript according to your hints.
The following is a detailed description of the revised content of the manuscript.
Point 2: Grammar problems have been marked in the MS
Response 2: Dear reviewer, thank you for your carefully correction. We have corrected all grammatical problems in this manuscript according to your annotations in the MS, including deleting superfluous words, changing punctuation marks, adding spaces, rewriting sentences, adding prepositions, etc., all of which are marked with a yellow background in MS.
Point 3: I would like to request authors to replace nitrogen to N as they used it many times.
Response 3: Dear reviewer, thank you for for this comment. We have replaced nitrogen to N in this MS.
Point 4: Figure text font style should be same as article text. Add tickmark in both axises.
Response 4: Dear reviewer, thank you for for this comment. We have changed all figures to ensure the text font style is the same as article text, and also added tickmark in both axises.
Point 5: 2.2. Experimental Design and Management: The highlighted part is contradictory. Please correct.
Response 5: Dear reviewer, thank you for for this comment. We have corrected this sentense as follows:
The study conducted a single-factor randomized block test consisting of nine treatments. These treatments were named as follows: CK0 (control, no fertilization), CK1 (control, one-time fertilization of common compound fertilizer with a conventional N application rate of 187.5 kg ha-1 in wheat and 225 kg ha-1 in maize), CK2 (comparison, which involved using common compound fertilizer as base fertilizer and urea topdressing). The N application amount was conventional, with 187.5 kg ha-1 in wheat and 225 kg ha-1 in maize.
Point 6: 5: 2.2. Experimental Design and Management: The treatment is confusing to the readers. Please explain in an easy way.
Response 6: Dear reviewer, thank you for for this comment. We have explained this paragraph more simply.
jf-20% and jf-30% (stir and mix biostimulant and common compound fertilizers. Compared to the common N application rate, it was reduced by 20% and 30%, respectively), jn-20%, jn-30% and jn-40% (use biostimulant chelated urea-formaldehyde fertilizer. Compared with the common N application rate, it was reduced by 20%, 30% and 40%, respectively), jny-27% (use jn-40% treatment and mixed with organic fertilizer, compared with the common N application rate, it was reduced by 27%).
Point 7: Clarify what is nutrient organ.
Response 7: Dear reviewer, thank you for for this comment. We have explained the nutrient organ in the MS.
nutrient organs (Stem and leaf)
Point 8: Simplify the equation like, Economic benefits = Grain yield benefits - Cost of production (.......)
Response 8: Dear reviewer, thank you for for this comment. We have simplified this equation:
Economic benefits = Grain yield benefits - Cost of production (Fertilizer seed pes-ticide costs, land rent and field management costs and seed and pesticide costs)
Point 9: I would like to request the authors to add n (number of replication) and SD/SE values
Response 9: Dear reviewer, thank you for for this comment. We have added the n in the annotations of the individual diagrams and the SD values in each Table.
Point 9: Caption should be below the figure.
Response 9: Dear reviewer, thank you for for this comment. We have adjusted the position of the image to ensure that the caption is below the image.
We appreciate your warm work earnestly and hope that the corrections will meet with approval. Once again, thank you very much for your comments and suggestions.
Yours sincerely
Junji Li
Corresponding author: Gaoqiong Fan
Email: fangao20056@126.com

Reviewer 3 Report
The manuscript assessed the overall effects of reducing nitrogen levels in conjunction with 17 the application of biostimulant on the yield formation and nitrogen utilization of wheat and maize. The findings of the research provide essential information and knowledge to the agricultural sector and relevant organizations. Generally, the manuscript is well-written, however, some sections of the manuscript require amendments/ improvements in order to enhance the quality of the manuscript. The comments are as follows:
Abstract:
The authors have provided a clear research background, problem statement, and objective in the Abstract section. However, it would be better if the authors elaborate a bit more on the methodology, for example description of the field experiment, and which parameters were assessed.
Introduction:
The authors had provided a very clear background of study, problem statement, and objective of the study in the Introduction section.
Materials and Methods:
1). In the Experimental Design and Management section, suggest the authors to provide more information on the plot preparation and descriptions (for example, size of the plots, how long the planting duration for one cycle, how did the fertilizers be applied, how many plants were planted in one plot, how did you carry out the daily maintenance of the plants and so on).
2). Statistical analysis information is missing in the Materials and Methods section. The authors need to provide a description of the statistical analysis and software being used to analyze the data.
Results and Discussion:
The results were well presented, elaborated, and discussed.
Conclusion:
The findings of the study were well concluded. Suggest the authors to provide a recommendation for future research at the end of the Conclusion section.
English Language:
English language in this manuscript is fine and extensive English editing is not required.
Author Response
Response to Reviewer 3 Comments
Dear reviewer 3:
Thank you for your comments concerning our manuscript entitled “Comprehensive Effects of Nitrogen Reduction Combined with Biostimulant on Nitrogen use efficiency and Yield of Winter Wheat-Summer Maize Rotation System (agronomy-2595005)”. These all comments are valuable and very helpful for revising and improving our MS, as well as are an important guidance to our researches. We have carefully checked and prepared the MS according to your suggestions. We hope that the revised manuscript will be according to your concerns. Revised portions are highlighted in yellow color of manuscript. The corrections in the paper and responses to the reviewer’s comments are as following:
Point 1: The manuscript assessed the overall effects of reducing nitrogen levels in conjunction with 17 the application of biostimulant on the yield formation and nitrogen utilization of wheat and maize. The findings of the research provide essential information and knowledge to the agricultural sector and relevant organizations. Generally, the manuscript is well-written, however, some sections of the manuscript require amendments/ improvements in order to enhance the quality of the manuscript.
Response 1: Dear reviewer, thank you for your careful review and appreciations. In addition, we have modified the MS in response to your suggestions to further improve its quality.
Point 2: Abstract: The authors have provided a clear research background, problem statement, and objective in the Abstract section. However, it would be better if the authors elaborate a bit more on the methodology, for example description of the field experiment, and which parameters were assessed.
Response 2: Dear reviewer, thank you for this comment. We have elaborate description of the field experiment and which parameters were assessed as follows:
Therefore, based on the winter wheat-summer maize rotation system in the modern R&D base of Sichuan Agricultural University, soil enzyme activities, soil inorganic nitrogen dynamic content, crop nitrogen accumulation and transportatin, crop yields and composition were determined.
Point 3: Introduction: The authors had provided a very clear background of study, problem statement, and objective of the study in the Introduction section.
Response 3: Dear reviewer, thank you for your careful review and appreciations.
Point 4: Materials and Methods:
1). In the Experimental Design and Management section, suggest the authors to provide more information on the plot preparation and descriptions (for example, size of the plots, how long the planting duration for one cycle, how did the fertilizers be applied, how many plants were planted in one plot, how did you carry out the daily maintenance of the plants and so on).
Response 4: Dear reviewer, thank you for this comment. We have provided the information on the plot preparation and descriptions as follows:
The winter wheat-summer maize crop rotation model was implemented in this study, with the previous crop being wheat. Following the harvest, all straw was removed from the experimental plot to ensure that soil fertility was not compromised. The plot was then tilled and sorted. Each plot in the experiment had an area of 21 m2 (4.2 × 5 m), and there were 3 replicates per treatment, resulting in a total of 27 plots. Wheat was sown in rows and covered with soil after sowing. The row spacing of each plot was 20 cm, with a total of 21 rows. The initial seedling density was set at 2.0 × 106 plants ha-1. Sowing dates for wheat were 27 October 2021 and 24 October 2022, with harvest dates of 8 May 2022 and 4 May 2023, respectively. The topdressing dates for CK2 were November 24, 2022 and November 22, 2023, respectively. For maize, it was sown in acupoints with a row spacing of 70cm and a hole spacing of 25cm. The initial seedling density at the four-leaf stage was 5.72 × 104 plants ha-1. Maize was sown on 13 May 2021 and 19 May 2022, respectively, and harvested on 2 September 2021 and 11 September 2022, respectively. Except urea as topdressing, other fertilizers were used as base fertilizer after crop seeding. The field management practices for wheat and maize were consistent with those followed in local high-yield fields.
Point 5: Materials and Methods:
2). Statistical analysis information is missing in the Materials and Methods section. The authors need to provide a description of the statistical analysis and software being used to analyze the data.
Response 5: Dear reviewer, thank you for this comment. We have provided the information of Statistical analysis as follows:
2.4. Date Analysis
The Excel2010 software was used to collate the experimental data, the IBM SPSS Statistics 22.0 software was used for variance analysis (ANOVA). The least significant difference (LSD) method was used for significance test (p < 0.05). The image visualiza-tion was performed by Origin 2021.
Point 6: Results and Discussion: The results were well presented, elaborated, and discussed.
Response 6: Dear reviewer, thank you for your careful review and appreciations.
Point 7: Conclusion: The findings of the study were well concluded. Suggest the authors to provide a recommendation for future research at the end of the Conclusion section.
Response 7: Dear reviewer, thank you for your careful review and appreciations. And we have provided the recommendation for future research at the end of the Conclusion section as follows:
Further research is required to investigate the nitrogen reduction effect of biostimulants combined with compound fertilizers in different soil textures and regions.
Point 8: English Language: English language in this manuscript is fine and extensive English editing is not required.
Response 8: Dear reviewer, thank you for your careful review and appreciations.
We appreciate your warm work earnestly and hope that the corrections will meet with approval. Once again, thank you very much for your comments and suggestions.
Yours sincerely
Junji Li
Corresponding author: Gaoqiong Fan
Email: fangao20056@126.com
